# Hydroponic Lettuce Cultivation Using Organic Nutrient Solution from Aerobic Digested Aquacultural Sludge

**Maha Ezziddine [1,*], Helge Liltved [1] and Randi Seljåsen [2]**

[1] Department of Engineering Sciences, University of Agder, 4898 Grimstad, Norway; helge.liltved@uia.no
[2] Norwegian Institute of Bioeconomy Research, 4886 Grimstad, Norway; Randi.Seljaasen@nibio.no
* Correspondence: maha.ezziddine@uia.no

**Abstract:** The aim of this study was to demonstrate how aquacultural sludge can be processed and utilized as an organic nutrient solution (ONS) for hydroponic lettuce production. By using a previous developed method, approximately 80% of the processed sludge was reclaimed as a clear, nutrient-rich solution. The performance of the recovered nutrient solution on lettuce growth was assessed in a nutrient film hydroponic system. The results were compared to the results obtained using a conventional nutrient solution (CNS). Yield, fresh weight, water consumption, and nutrient and heavy metal content in leaf tissue were measured. In spite of a 16% lower average fresh weight obtained in ONS compared to the weight obtained in CNS, there was no statistical difference of the yield of lettuce among the two nutrient solutions. After the cultivation period, 90% of the lettuce heads grown in ONS exceeded the marked weight of 150 g. Foliar analysis revealed a similar or higher content of all nutrients, except of magnesium and molybdenum in the leaves of lettuce grown in the ONS compared to lettuce grown in the CNS. This study shows that nutrients recovered from aquacultural sludge can be utilized as fertilizer, thereby reducing the dependency on mineral fertilizer in hydroponic and aquaponic systems.

**Keywords:** organic fertilizer; nutrient recovery; organic nutrient solution; nutrient film technique; aquacultureal sludge; heavy metals

## 1. Introduction

In hydroponic systems, plants are grown without soil. In ordinary growth systems, soil supports plants' roots and provides water, nutrients, and oxygen to these roots. On the contrary, in hydroponic systems, plant roots are supported by an inert medium in net pots while water and nutrients are delivered via nutrient solution [1]. There are three commonly used hydroponic technologies: media bed hydroponic, nutrient film technique (NFT) and deep-water culture (DWC) technologies. NFT was the hydroponic technology used in this study. NFT systems have been widely discussed and tested since developments in hydroponics in the 1970s [2]. In NFT, seedlings are normally placed in net pots filled with substrates. Net pots are then placed in a trough, channel, gully, or pipe with holes for the net pots. This provides physical support for the plants. Inside the trough, a film of 1–2 cm of nutrient solution flows along the bottom [3]. Part of the roots develops in the film of the nutrient solution while the other part is suspended in the air above, which ensures that the roots receive sufficient quantities of oxygen and have enough air exchange surface. Pipes are usually made from PVC (since it is inexpensive) and have a white color which reflects light and avoids excessive heating. Pipes can be round or rectangular. Rectangular pipes with a width larger than their height allow for a much larger surface area of the nutrient solution, which increases nutrient uptake and plant growth [3]. Pumps are used to circulate the nutrient solution from the reservoir to the pipes which, preferably, should be positioned on a slope to facilitate nutrient solution flow along the pipe.

Unlike soil culture, which is inefficient in water and nutrient reuse, closed hydroponic systems conserve water and nutrients through the recirculation of the nutrient solution [4].

While hydroponic systems are considered to represent a sustainable method for grow plants [5], the nutrient solution used in hydroponic systems is based on chemical fertilizers which are mined from scarce and non-renewable resources [6]. Recently, there has been an increased interest in organic hydroponics, as the market for organic food continues to grow [7]. Some studies have reported the possibility of growing vegetables using an organic nutrient solution (ONS) [4,8,9].

Several organic fertilizers have been tested in hydroponic systems such as bonito soup, rapeseed oil cake, corn oil cake, fish meal, dried brewer's yeast, fermented tomato foliage, sea weed, and vermicaste derived solutions [5,10]. Phibunwatthanawong and Riddech [9] produced a liquid organic fertilizer for hydroponic systems from fermented molasses, distillery slop, and sugarcane leaves which had similar growth effect as chemical fertilizers.

However, some challenges in using ONS in hydroponics have been reported, such as a lower growth rate compared to conventional hydroponics and difficulties managing both pH and EC [4,5,7]. Atkin and Nichols [5] showed that lettuce grown in NFT with a conventional nutrient solution weighed 200% more than lettuce grown in ONS derived from liquid fish and liquid seaweed. Similarly, Williams and Nelson [11] reported that the fresh and dry weights of lettuce gown in NFT were lower in organically-fertilized cultivation compared to conventionally-fertilized cultivation.

The direct use of organic fertilizers in hydroponic systems may inhibit plant growth due to high biological oxygen demand in the root zone caused by the presence of dissolved organic carbon compounds [12]. Additionally, most of the nutrients in organic sources, such as waste material from the agricultural and aquacultural industry, are not in ionic forms and, hence, are not directly available for plants. For optimizing the utilization of organic waste for hydroponic plant growth, a solubilization step is required to break down organic matter and mobilize nutrients.

Aquacultural sludge contains high amounts of nutrients, the majority of which are bound to organic matter [13,14]. In order to increase the ratio of plants' available nutrients in aquacultural sludge, aerobic digestion (AD) has been used to solubilize nutrients from organic compounds while breaking them down [15–17]. The solubilization step should be followed by a solid precipitation step in order to obtain a good quality solids-free phase, in compliance with water quality guidelines for hydroponic and aquaponic systems. Chitosan, which is a natural polymer made from shrimp and crab shells, has been shown to be an effective flocculant for solids precipitation from aquacultural sludge [16]. Unlike other metal-based coagulants and synthetic polymers for the flocculation and precipitation of solids, chitosan does not combine chemically with dissolved phosphate and has no health concerns [18].

In addition to nutrients for plant growth, aquacultural sludge may also contain heavy metals which can limit the use of sludge as fertilizer. Heavy metals may accumulate to levels exceeding permissible content in crops for human consumption. In recirculating aquacultural systems (RAS), heavy metals may enter with the feed, could leach from pipes and fittings, or could be carried into the system with the make-up water [19].

The aim of this study was to demonstrate how aquacultural waste can be processed and utilized as a valuable nutrient solution for hydroponic growth. Aquacultural sludge was processed by AD, and clarified by chitosan flocculation and sedimentation, as described by Ezziddine et al., 2020 [16]. The performance of the recovered ONS solution on lettuce growth was assessed by cultivation experiments where the fresh weight of lettuce, yield, water consumption, and nutrient and heavy metal content in the leaf tissue were measured. The results were compared to the results obtained by use of a conventional mineral nutrient solution.

## 2. Materials and Methods

### 2.1. The Aquaponic Facility and Sludge Collection

For this study, 120 L of sludge was collected from an aquaponic research facility at the Norwegian Institute of Bioeconomy Research (NIBIO, Landvik, Norway). The aquaponic

facility is a closed system based on RAS technology connected to tanks for the production of vegetables and consists of two separate identical test units (unit A and unit B). Figure 1 show a flowchart of one of the two units. The total water volume of each unit was 8 m$^3$, divided into two fish tanks with a volume of 1.2 m$^3$ each, and two plant compartments with a volume of 3 m$^3$ each. A swirl separator was mounted outside each fish tank to collect uneaten fish feed and feces directly from the outgoing water. A pumping sump for each unit provides water to the fish tanks, plant compartment, and water treatment system. The water treatment system consists of swirl separators, a combined particle and biofilter (Polygeyser Bead Filter, New Orleans, USA), a heating/cooling unit, an aeration unit, and a pH-control unit (Figure 1). The fish tanks were stocked with 530 brown trout (*Salmo trutta*) with an average weight of 97 g. The daily amount of feed given was 300 g, 75 g of which was distributed evenly to each tank by automatic feeders.

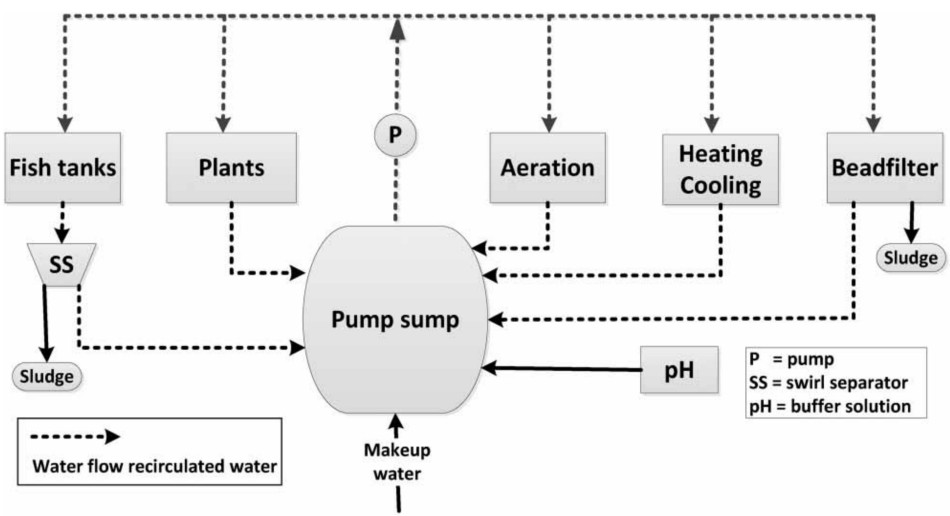

**Figure 1.** Flowchart of one of the two identical units of the aquaponic research facility showing water flows, water treatment, and sludge removal sites. The facility co-produced fish and vegetables with 100% recirculation of water.

The sludge which consisted of backwash-water from the bead filters and drainage from the swirl separators was transferred and mixed in a common aerated tank from where sludge used for this study was collected.

### 2.2. Aerobic Digestion and Solid Separation

For AD, the sludge was distributed to six 25 L polyethylene batch reactors covered with lids to minimize evaporation. During the four-week digestion period, each reactor was aerated at a continuous rate of 20 L min$^{-1}$, provided by diffusors and a common blower. The temperature was kept constant at 22 °C. The pH was measured daily in all reactors and maintained at approximately 7 by using solutions of HNO$_3$ (60%) and KOH (20%) for adjustment. The sludge was characterized before and after the four-week AD period. Representative 0.5 L grab samples were collected from each reactor and transferred to a common beaker. Well mixed duplicate composite samples for analysis were collected from the beakers. The samples were analyzed for total suspended solids (TSS), chemical oxygen demand (COD), biological oxygen demand (BOD), total nitrogen (TN), and total phosphorus (TP).

For removal of residual solids from the clear phase after AD, flocculation and sedimentation, using chitosan as coagulant, was conducted. According to experiences from previous research with similar sludge [16], a commercial chitosan (Kitoflokk High from Teta Vannrensing AS, Norway) at a dosage of 15 mg L$^{-1}$ was applied. The chitosan had a molecular weight of 161 kD and a degree of deacetylation of 80%. Chitosan stock solution (0.25%) was prepared by mixing 0.25 g of chitosan powder with 100 mL of distilled wa-

ter, and then adding drops of a 60% $HNO_3$ solution under continuous stirring until the chitosan was dissolved. The chitosan solution was added to the aerobic digested sludge during rapid mixing (400 rpm) and adjustment of the coagulation pH to approximately 6. After 60 s of rapid mixing, followed by 10 min of flocculation under slow stirring (30 rpm) and 20 min sedimentation, samples were collected from the clear phase for analysis of solubilization of nutrients during the four-week AD. 0.5 L grab samples were collected from each reactor both before the four-week AD period and after AD and solid separation. From the mixture of grab samples, composite samples were collected, and membrane were filtered by using Whatman GFC 0.45 μm filters prior to analysis of the following nutrients: ammonium ($NH_4$-N), nitrate ($NO_3$-N), phosphorous (P), potassium (K), calcium (Ca), sulfur (S), magnesium (Mg), copper (Cu), zinc (Zn), manganese (Mn), molybdenum (Mo), boron (B), and iron (Fe). A sample from the concentrated sludge (precipitated solids) was also taken for TSS, BOD, COD, TP, and TN analysis. The clear water was stocked in the fridge for use as ONS for hydroponic lettuce growth.

### 2.3. Lettuce Growth

The growth performance of lettuce in the nutrient solution prepared from aerobic digested aquacultural sludge was compared to growth in conventional mineral nutrient solution. The experiments were carried out in a growth room located at the University of Agder in Norway. The study with the ONS was carried out from the 7 October to the 19 November 2020, while the study with CNS was carried out from 27 September to the 4 December 2018. The studies were conducted in triplicate experimental set-ups under identical conditions. The temperature of the growth room was in the range of 21–24 °C, the $CO_2$ concentration was in the range 450–580 ppm, and the relative humidity was 40–55%.

The growth systems consisted of a seeding system and a closed loop NFT system. In the seeding system, seeds of *Lactuca sativa* L. (Batavia-type, cv. "Partition") from LOG AS, Norway, were seeded in Grodan rockwool cubes ($36 \times 36 \times 40$ mm) and placed in a tray filled with nutrient solution and illuminated with LED-light. After two weeks in the seeding system, the seedlings in the rockwool cubes were inserted into net pots and transplanted to the NFT-system which consisted of three identical parallel units. Each unit consisted of a rectangular PVC-pipe (240 cm long, 10 cm width and 50 cm height) with 12 holes (each with a diameter of 45 mm), and a 20 L plastic container filled with nutrient solution. From the plastic container, the nutrient solution was supplied intermittent (30 min on/off cycles) to the PVC-pipe by a submerged pump at a flow rate of 3.5 L min$^{-1}$. The hydroponic system was illuminated 18 h per day with LED-light transmitting photosynthetically active radiation (PAR) of a photon flux density of 220 μmol m$^{-2}$ s$^{-1}$.

The pH and EC of the three parallel systems were monitored and adjusted every second day. The target values of pH and EC were 6.0 and 1200 μS cm$^{-1}$, respectively. Soluble N and P in the nutrient solution were measured two times per week using a Hach DR3900 spectrophotometer (Loveland, CO, USA). After four weeks in the NFT-systems, lettuce was harvested and weighed. Leaf tissue samples were collected for nutrients content and heavy metal analysis. Samples from the remaining nutrient solution were also taken for soluble nutrient analysis.

### 2.4. Analytical Methods

TSS, TN, TP, BOD, and COD were determined by Eurofins laboratory, Norway. For TP and TN, the nutrients were dissolved by nitric acid microwave extraction and analyzed using ICP-OES according to European standards (DIN EN ISO 11885). Soluble nutrients (P, K, Cu, Zn, S, Mn, Mg, Ca, Mo, B, Fe) were analyzed using inductively coupled plasma optical emission spectrometry (ICP-HSP) according to accredited standards by the Eurofins laboratory, Netherlands. Leaf tissue was collected from different parts of the lettuces to avoid biases due to uneven nutrient distribution within the plants. The nutrients and metals were dissolved by nitric acid microwave extraction and analyzed by inductively coupled plasma atomic emission spectrometry (ICP-OES) according to European standards

(DIN EN ISO 11885). Operational parameters including turbidity, pH, EC, and TSS of the nutrient solution were measured in the laboratory of University of Agder according to Norwegian and European standards. TSS was measured using pre-weighed 0.45 μm Whatman GF/C glass microfiber filters. Turbidity was determined using a calibrated Hach 2100Q turbidimeter (Loveland, CO, USA), while pH and EC were measured using a calibrated Hach HQ40d instrument with standard pH and EC sensors. All samples were stored in a refrigerator at 4 °C and sent immediately after sampling in insulated cooler bags to the laboratory.

### 2.5. Data Analysis and Statistics

Data were processed in Microsoft Excel (2016) and were subjected to analysis of variance (ANOVA) using SPSS Version 25. Mean differences were determined by Tukey's honestly significant difference (HSD) test at $p < 0.05$. Mean values with standard deviations are presented. Where standard deviation bars are not shown on the graphs, they do not extend beyond the dimensions of the symbols.

## 3. Results and Discussion

### 3.1. Sludge Characterisation before and after AD

TSS, BOD, COD and nutrient content in aquacultural sludge can vary from one facility to another depending on several factors including fish species, fish density, fish age, fish feed, feeding management, flow regulation, and water treatment (including solid separation technology and sludge handling). The sludge used in this study had a relatively low TSS content of 1.1 g L$^{-1}$ (Table 1). The suspended solids were generated from fish feces, uneaten food, and biofloc (suspended flocs formed by different microorganisms that adhere to an organic matrix [20]). The BOD$_5$ concentration of the sludge was 145 mg L$^{-1}$ while the COD concentration was approximately twelve times higher, indicating a high share of recalcitrant organic matter which is not easily degradable by aerobic microorganisms.

**Table 1.** Characteristics of aquacultural sludge before and after aerobic digestion. Mean values are given.

| Parameter | Before AD | After AD | Remaining Concentrated Sludge after AD and Solid Separation |
|---|---|---|---|
| Volume of sludge (L) | 120 | 101 | 5 out of the 101 |
| pH | 7.1 | 7.0 | 7.0 |
| TSS (g L$^{-1}$) | 1.10 (132.0) | 0.46 (46.5) | 10.00 (50.0) |
| BOD$_5$ (mg L$^{-1}$) | 145 (17.4) | 16 (1.6) | - |
| COD (mg L-1) | 1750 (210) | 390 (39.4) | 7900 (39.5) |
| Total P (TP) (mg L$^{-1}$) | 92 (11.0) | 93 (9.4) | 560 (2.8) |
| Total N (TN) (mg L$^{-1}$) | 77.5 (9.3) | 120 (12.1) | 170 (0.9) |

Numbers in bracket represent the mass (g) of the compounds (volume of sludge multiplied by concentration).

During the four-week period of aerobic degradation, the sludge volume was reduced from 120 L to 101 L due to evaporation (Table 1). To account for the volume reduction of 16%, all of the following calculations were done on a mass basis. As shown in Table 1, TSS was reduced from 1.10 g L$^{-1}$ (which corresponds to a mass of 132.0 g) to 0.46 g L$^{-1}$ (which corresponds to 46.5 g), which corresponds to a reduction of 64.8% during AD. Also, the majority of organic matter in the sludge was oxidized, with BOD and COD reductions of 90.8% and 81.2%, respectively. There was only a small decrease in TP, while an increase in TN of 30.1% was indicated. Such an increase has also been observed during AD in other studies [16], and can partly be explained by nitrogen fixation. It has been reported that some free-living heterotrophs can fix significant levels of nitrogen without the direct interaction with other organisms [21]. The remaining solids in the aerobic digested sludge were precipitated using chitosan as a coagulant. After solid removal by coagulation and 15 min of sedimentation, 96L of the 101 L was recovered as a nutrient rich clear phase for further studies, while 5 L remained as a concentrated sludge (Table 1). The solid content of the clear phase was as low as 6.7 mg L$^{-1}$, which corresponded to 99.0% solid removal by



coagulation and sedimentation. The turbidity of the clear phase was 1.64 FNU, which is in the range of potable water quality.

### 3.2. Nutrient Mobilization during Aerobic Digestion

In Table 2, the concentrations of soluble macro and micronutrients in the sludge before AD, and in the clear phase after AD and solid separation, are shown. By comparing the masses of soluble P and N ($NO_3$-N) before AD (Table 2) by the masses of total P and N before AD (Table 1), it is indicated that substantial amounts of these nutrients were associated with solids initially (75.5% of P and 47.7% of N). After AD, all particulate N were mobilized and 62.8% of the total phosphorus was in soluble form. As shown in Table 2, there was a 2.6-fold elevation in the concentration of soluble P, while the corresponding increase for soluble N ($NO_3$-N) was 3.0-fold. Calculated increases based on mass gave 16% lower values due to the volume decrease by evaporation during AD. It was further shown that all other concentrations of dissolved macro- and micronutrients increased during AD, except Mn. Of the macronutrients, K, Ca, Mg, and S exhibited elevations in concentrations by factors of 1.8, 1.9, 1.8, and 1.4. Among the micronutrients, Fe, B, Cu, and Zn were raised by factors of 1.1, 2.0, 1.5, and 1.5, while there was a decrease in Mn-concentration.

**Table 2.** Concentrations of soluble nutrients in the sludge before and after aerobic digestion. Mean values and standard deviations (STD) are given.

| Parameter | Before AD | | After AD and Solids Separation | | Standard Recommended Range [2] |
|---|---|---|---|---|---|
| | Mean | STD | Mean | STD | |
| Soluble macronutrients | | | | | |
| $NH_4$-N (mg L$^{-1}$) | <1.5 | | <1.5 | | 100 to 200 |
| $NO_3$-N (mg L$^{-1}$) | 40.5 (4.9) | 1.5 | 121.7 (12.3) | 3.09 | |
| P (mg L$^{-1}$) | 22.5 (2.7) | 0.5 | 58.0 (5.9) | 3.74 | 15 to 90 |
| K (mg L$^{-1}$) | 104.0 | 2 | 185.3 | 12.36 | 80 to 350 |
| Mg (mg L$^{-1}$) | 9.7 | 0 | 17.0 | 0 | 26 to 96 |
| Ca (mg L$^{-1}$) | 64.0 | 0 | 124.0 | 6.53 | 122 to 220 |
| S (mg L$^{-1}$) | 13.0 | 0 | 18.0 | 1.41 | |
| Soluble micronutrients | | | | | |
| B (mg L$^{-1}$) | 0.025 | 0.005 | 0.050 | | 0.14 to 1.5 |
| Cu (mg L$^{-1}$) | 0.02 | 0 | 0.03 | 0 | 0.07 to 0.1 |
| Mn (mg L$^{-1}$) | 0.085 | 0.005 | 0.020 | 0 | 0.5 to 1 |
| Mo (mg L$^{-1}$) | <0.01 | | <0.01 | | 0.05 to 0.06 |
| Zn (mg L$^{-1}$) | 0.56 | 0 | 0.83 | 0.03 | 0.5 to 2.5 |
| Fe (mg L$^{-1}$) | 1.15 | 0.025 | 1.24 | 0.05 | 4 to 10 |
| Others | | | | | |
| Na (mg L$^{-1}$) | 18 | 0 | 22.3 | 0.94 | |
| Cl (mg L$^{-1}$) | 26.5 | 1.5 | 41.3 | 4.64 | |

Numbers in bracket represent the mass (g) of the compounds (volume of sludge multiplied by concentration).

Most of the macronutrients after AD were in concentration ranges recommended for hydroponic growth systems, except for Mg. Also, some of the micronutrients, e.g., B and Mn, were lower in concentration than recommended. Despite these shortages in some nutrients, the results indicate that aquacultural sludge, after aerobic digestion, has the potential to fulfil the requirements of a nutrient solution for use in hydroponic systems, in line with the previous study conducted by Ezziddine et al. (2020) [16].

### 3.3. Comparative Lettuce Growth Studies

The chemical composition of the CNS and the ONS used in the comparative growth study are presented in Table 3. As shown, there are some differences. Among the macronu-trients, the concentrations of N, P, K and Ca were higher in the ONS than in the CNS,

while Mg and S were slightly lower. Regarding the micronutrients, the concentrations were generally lower in the ONS, except for Zn which was present at a concentration of 0.83 mg $L^{-1}$.

**Table 3.** Soluble nutrient content of the conventional mineral nutrient solution and the organic nutrient solution.

| Parameter | Soluble Concentration (mg $L^{-1}$) | | | | | | | | | | | | |
|---|---|---|---|---|---|---|---|---|---|---|---|---|---|
| | $NO_3$-N | $NH_4$-N | P | K | Ca | Mg | S | Zn | B | Cu | Fe | Mn | Mo |
| Applied ONS | 121.7 | <1.5 | 58 | 185 | 124 | 17 | 18 | 0.83 | 0.05 | 0.03 | 1.2 | 0.02 | <0.01 |
| Applied CNS | 107 | 4.0 | 23 | 140 | 94 | 23 | 23 | 0.26 | 0.19 | 0.07 | 1.7 | 0.42 | 0.04 |

3.3.1. pH and EC Values during Growth in the NFT-System

At the beginning of the growth studies with lettuce, the pH of the ONS was 6.0, while that of the CNS was 6.2, as shown in Figure 2a. The pH was adjusted every second day to the target pH-value of 6.0. The data points represent the mean pH value of the triplicate NFT-units just before pH adjustment. For the ONS, an increase in pH over time was observed, despite the pH-adjustment to 6.0 every second day. The pH increase was more severe at the end of the experimental period. Unlike the ONS, the pH of the CNS tended to decrease below 6, down to 5.5, during the early phase of the growth period. This was in spite of pH adjustment with KOH. In the literature, pH increase and decrease during growth have been explained by the $NH_4$-N to $NO_3$-N ratio of the nutrient solution [2]. By elevating the ratio (increasing $NH_4$-N), the pH tends to decrease in the nutrient solution. This could be some of the explanation of the observed differences in pH-behavior in our case, since $NH_4$-N contributed to 4% of the dissolved N in the CNS, while $NH_4$-N was below the detection limit of 1.5 mg $L^{-1}$ in the ONS.

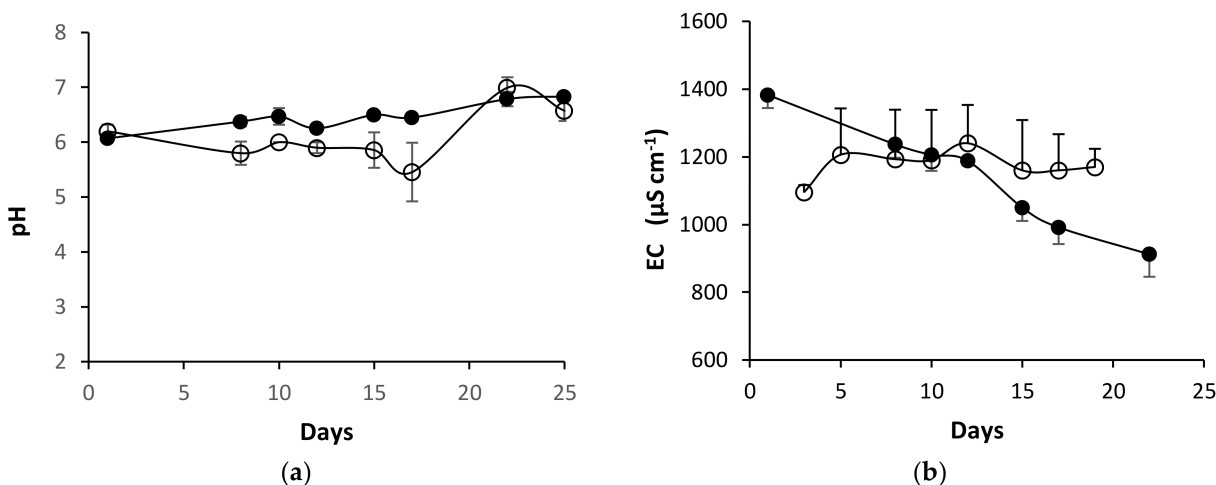

**Figure 2.** Variation in (**a**) pH and (**b**) EC in the organic (●) and conventional (○) nutrient solution over the 25 days experimental period.

The recovered ONS was used undiluted with initial EC of 1383 μS $cm^{-1}$ (Figure 2b). After 10 days, the EC value was reduced to 1206 μS $cm^{-1}$. Volumes of ONS was added to maintain the volume of 20 L and EC at approximately 1200 μS $cm^{-1}$, which was the target EC-value. However, EC continued to decrease despite nutrient additions, and was down to 912 μS $cm^{-1}$ at the end of the experiment.

The EC of the CNS was easier to maintain at approximately 1200 μS $cm^{-1}$, and remained quite stable. However, the average concentrations of the ONS and the CNS throughout the experimental periods were quite similar (1140 μS $cm^{-1}$ and 1170 μS $cm^{-1}$, respectively).

### 3.3.2. PO$_4$-P and NO$_3$-N Concentrations of the ONS during Growth in the NFT-System

The concentration of NO$_3$-N and PO$_4$-P in the ONS during lettuce growth in the NFT-system are shown in Figure 3. At the beginning of the cultivation period, NO$_3$-N concentration decreased significantly from 120 mgL$^{-1}$ to 63 mgL$^{-1}$. Then, it continued to decrease slowly until the end of the experimental period to a final concentration of 48 mgL$^{-1}$.

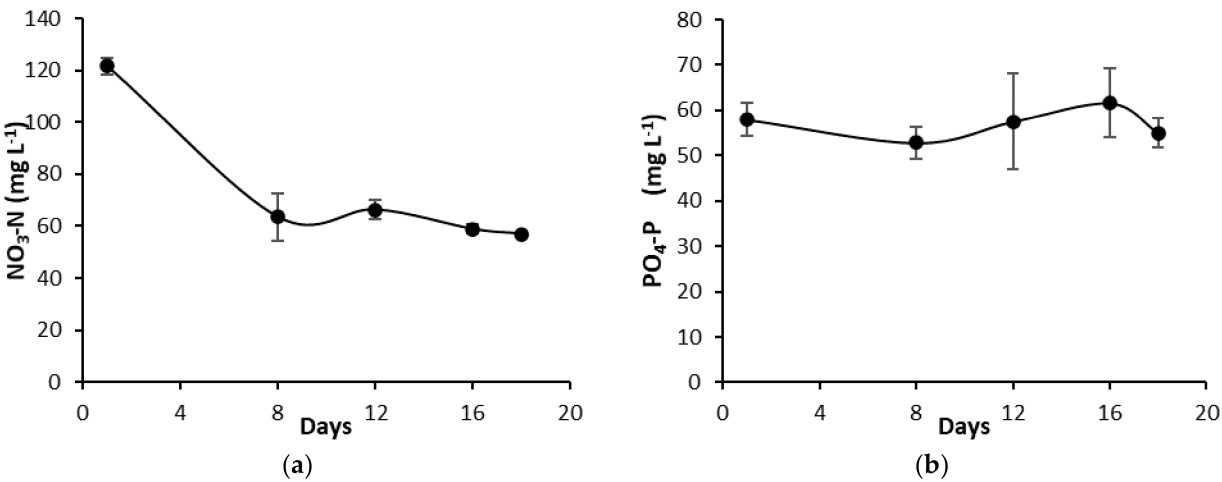

**Figure 3.** Concentration of (**a**) NO$_3$-N and (**b**) PO$_4$-P in the organic nutrient solution over the experimental period. Statistically significant differences are indicated by different letters.

The concentration of PO$_4$-P was almost stable over the experimental period in the range of 53–62 mg L$^{-1}$. No statistically significant differences were detected between individuals PO$_4$-P concentrations.

### 3.3.3. Effect of ONS versus CNS on Lettuce Growth and Yield

Shoot fresh weights of the harvested lettuce yields after five weeks of growth (two weeks in the seedling system and three weeks in the NFT-system) in the two different nutrient solutions are presented in the Figure 4 as box plots. The solid horizontal lines from bottom to top indicate the minimum, first quartile, median value, third quartile, and maximum values. The dots above and below represent outliers and the cross represents the average values. The lettuce grown in the ONS had equal median and average shoot fresh weights (203 g). This means there were similar numbers of plants with weights in the lower range as in the higher range. Weights between the first and the third quartile were from 165 g to 224 g. The lowest shoot fresh weight was 120 g while the highest was 270 g. One lettuce head weighing 345 g was considered to be statistically different and did not fit the fresh weight of shoots set but was still included in the statistical analysis.

The lettuces grown in the CNS had an average shoot weight of 243 g which was slightly higher than the median value of 238 g. This means that most of plants were in the lower range of the yield values. Values between the first and the third quartile were in the range from 210 g to 261 g. The minimum shoot fresh weight was 196 g and the maximum was 322 g. The shoot fresh weights of two lettuce heads (119 and 357 g) were considered as outliers but still included in the analysis.

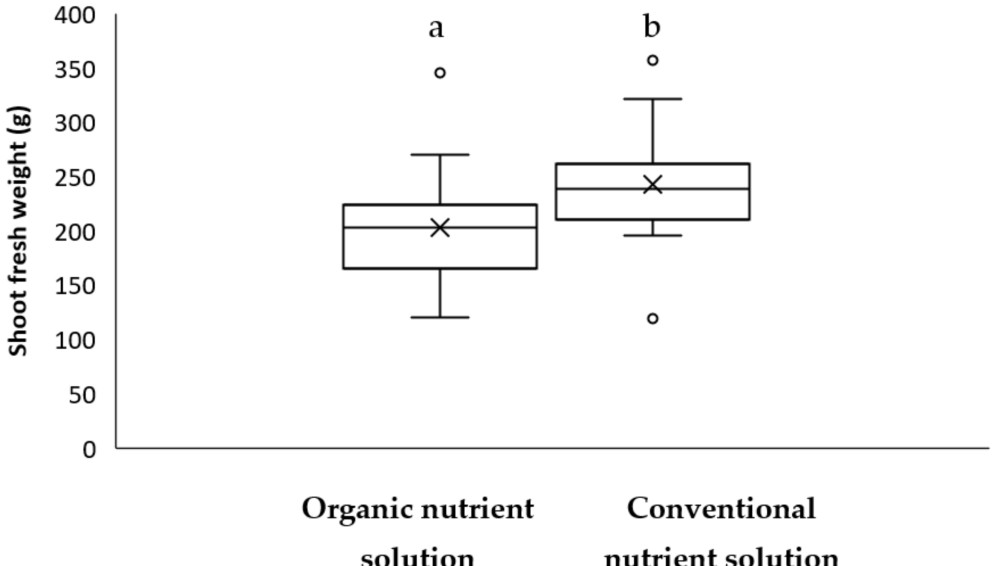

**Figure 4.** Shoot fresh weight of lettuce grown in organic nutrient solution and in conventional mineral nutrient solution. Different letters indicate statistically significant differences.

The average fresh weight obtained in ONS is 16% lower than the average weight obtained in CNS. Nonetheless, 90% of the lettuces grown in the ONS had a shoot fresh weight substantially higher than the marked weight of commercial lettuce heads (150 g). The shoot fresh weights of the lettuces grown in ONS had larger variance than those of the CNS, as reflected by the differences in the length of the boxes (Figure 4).

In Table 4, the nutrient content in leaves of lettuce grown in the ONS and in the CNS is shown. The lettuce grown in the ONS had a significantly higher concentrations of P, K, Ca, S, and B ($p < 0.05$), and significantly lower concentrations of Mg and Mo ($p < 0.05$) compared to the lettuce grown in the CNS. These differences were also reflected in the nutrient solution concentrations. The ONS was higher in concentration for all these elements, except for S and B (Table 3). In particular, the average concentrations of K and Ca in leaves of lettuce grown in ONS were 2.5 and 2.0-times higher than those grown in conventional solution, respectively. There were no significant differences in N, Cu, Mn, Zn, and Fe concentrations in leaves of lettuce from the two nutrient solutions. The low content of Mg in lettuce grown in the ONS can also be explained by high concentration of K and Ca. As reported by Senbayram et al. (2015), Ca and K in excess may interfere with Mg-uptake, which is known as nutrient antagonism [22].

The total yield of lettuce grown in the ONS was 3.87 kg m$^{-2}$ while the CNS resulted in slightly higher yield of 5.05 kg m$^{-2}$ (Figure 5). Interestingly, there was no significant difference between them, which means that the performance of the ONS regarding total yield was very close to the performance of CNS for hydroponic lettuce growth. Jordon et al. (2018) reported a yield of 2.18–2.58 kg m$^{-2}$ for hydroponic lettuce grown with CNS, which was lower than the yield obtained in ONS in this study [23]. The average water consumption of the lettuce grown in the CNS was 11.9 L kg$^{-1}$ while the lettuce grown in the ONS consumed 7.3 L kg$^{-1}$ which corresponds to a 61.4% reduction in the water consumption compared to the conventional lettuce (Figure 5). This reduction in water consumption can be explained by higher concentration of Na and Cl in the ONS. The concentrations of Na and Cl were higher by factors of 3 and 25, respectively, in the ONS compared to the CNS. Many studies have also reported a decrease in water consumption in plants due to salinity increase in the nutrient solution [24].

**Table 4.** Nutrients content in leaves of lettuce grown in the organic nutrient solution and in the conventional nutrient solution. Mean values and standard deviations (STD) are given. Statistically significant differences are indicated by different letters.

| | Organic Nutrient Solution | | Conventional Nutrien Solution | | Statistical Significance |
|---|---|---|---|---|---|
| | Mean | STD | Mean | STD | |
| **Macronutrients (g/kg Dry Matter)** | | | | | |
| N | 53 [a] | 0 | 38.6 [a] | 23.2 | 1 |
| P | 10.26 [a] | 0.5 | 6.4 [b] | 0.5 | 0.002 |
| K | 125 [a] | 1.6 | 50 [b] | 19.8 | 0.006 |
| Mg | 1.7 [a] | 0.08 | 3.36 [b] | 0.33 | 0.002 |
| Ca | 30 [a] | 2.16 | 15 [b] | 14.14 | 0.001 |
| S | 3.43 [a] | 0.1 | 2.7 [b] | 0.2 | 0.012 |
| **Micronutrients (mg/kg Dry Matter)** | | | | | |
| B | 34 [a] | 2.16 | 22.33 [b] | 1.25 | 0.003 |
| Cu | 5.4 [a] | 1.2 | 6.56 [a] | 0.6 | 0.28 |
| Mn | 119 [a] | 22.5 | 143.3 [a] | 26.25 | 0.37 |
| Mo | 0.6 [a] | 0.04 | 1.43 [b] | 0.4 | 0.046 |
| Zn | 52 [a] | 21.23 | 50.3 [a] | 14.38 | 0.93 |
| Fe | 102 [a] | 13.95 | 130 [a] | 8.16 | 0.07 |

Equal letters (a-a) indicate no statistical differences among mean values, while unequal letters (a-b) indicate significant differences.

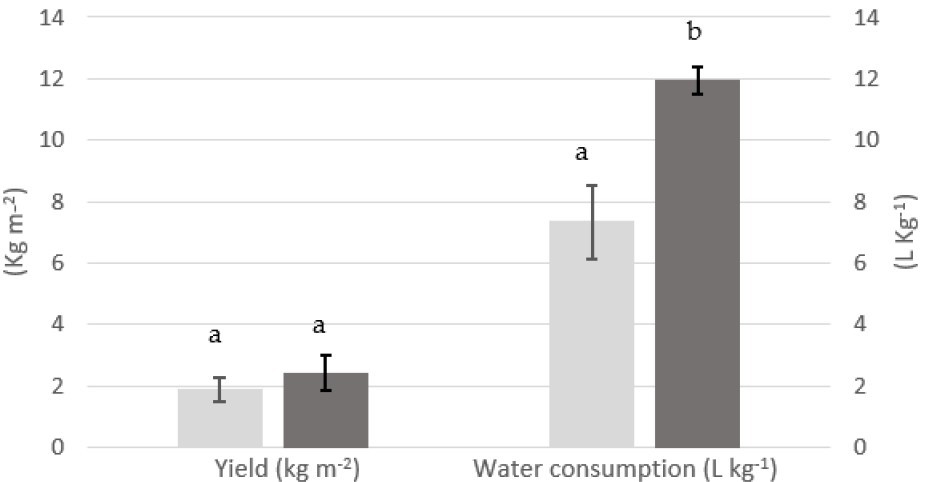

**Figure 5.** Yield of lettuce and water consumption by use of the organic and the conventional nutrient solution. Different letters indicate statistically significant differences.

*3.4. Heavy Metals Content in Lettuce Leaves*

As we have seen so far, aquacultural sludge is a valuable source of nutrients, which can replace mineral fertilizers. However, biosolids could possess high levels of heavy metals which can be toxic to plants, animals, and humans. Many authors have documented the absorption and phytotoxic effects of heavy metals on several crops [23].

Table 5 shows the concentrations of heavy metals (Pb, Cd, Ni, Zn, Cu) in the leaves of lettuce grown in the ONS and the maximum permissible concentrations in lettuce for human consumption set by the European Union and The Food and Agriculture Organization of the United Nations (FAO) [24].

**Table 5.** Heavy metals content in the leaves of lettuce grown in the organic nutrient solution and maximum permissible limit in lettuce for human consumption [25]. Mean values and standard deviations (STD) are given.

| | Heavy Metal Content in Leaves of Lettuce (mg kg$^{-1}$ Dry Weights) | | Maximum Permissible Limit (mg kg$^{-1}$ Dry Weights) |
|---|---|---|---|
| | Mean | STD | |
| Lead (Pb) | 0.21 | 0.06 | 0.3 |
| Cadmium (Cd) | 0.23 | 0.05 | 0.2 |
| Nickel (Ni) | <0.01 | - | 1.5 |
| Zinc (Zn) | 52 | 26.0 | 60–80 |
| Copper (Cu) | 5.4 | 1.21 | 40 |
| Arsene (As) | <0,04 | - | - |
| Chromium (Cr) | <1.0 | - | - |

As shown in Table 5, the content of the heavy metals Pb, Ni, Zn, and Cu in the leaves of lettuce grown in the ONS were all below the permissible limit for human consumption (Table 5), except for cadmium which was slightly above the maximum concentration. Eissa and Negim [25] also reported a high Cd content that exceed the permissible limit in lettuce grown on metal-contaminated soil. Zubillaga and Lavado [26] showed that lettuce grown in biosolids compost did not accumulate Cd concentration higher than the permissible limit. The content of lead and zinc were both close to the maximum concentrations. In order to safely use the ONS for lettuce growth, the concentration of heavy metals in the nutrient solution should be reduced. Different heavy metal removal techniques have been reported in literature including adsorption, membrane, chemical, electric, and photocatalytic based treatments [27]. In our case, we can increase the concentration of chitosan in the solid separation step as chitosan is considered to be a good metal chelator [28] which did not reduce the concentrations of metals essential for plant growth such as Mn and Fe [16]. Another startegy to reduce the heavy metal in the ONS is to choose fish feed with low metal content as in RAS, heavy metals may enter with the fish feed [29].

## 4. Conclusions

The current study demonstrated that ONS recovered from aquacultural sludge can be used for lettuce production in a hydroponic system after applying measures to control some heavy metals. The aquacultural sludge proved to be a good source of organic fertilizer. After aerobic digestion, approximately, 90% of the sludge was reclaimed as a clear, nutrient-rich solution that showed good performance on hydroponic lettuce growth. About 90% of the harvested lettuce heads reached or exceeded the marked size of commercial lettuce of 150 g after the five-week growth period. The yield of lettuce (kg m$^{-2}$) grown in ONS was comparable to the yield of lettuce grown in CNS. Except for Mg and Mn, comparable and even higher content of nutritionally minerals were found in the leaves of lettuce grown in organic fertilizer compared to the lettuce grown in conventional fertilizer. Interestingly, the consumption of water was 7.3 L per kg for organically-produced lettuce, and substantially lower than for conventionally-produced lettuce. In ONS grown lettuce, some heavy metals (Cd, Pb, and Zn) exceeded or were close to the maximum permissible concentrations in lettuce for human consumption, which indicates that the heavy metal content should be monitored closely when using aquacultural sludge as fertilizer for edible crop. This study suggests a method to recycle aquacultural sludge for use in organic hydroponics. It also recommends the use of aerobic degradation of sludge in aquaponic systems which can provide ONS to be reinserted into the water loop in coupled aquaponics or into the hydroponic system in decoupled aquaponics.

**Author Contributions:** Conceptualization, M.E. and H.L.; methodology, M.E. and H.L.; software, M.E.; validation, H.L.; formal analysis, M.E.; investigation, M.E. and H.L.; resources, H.L. and R.S.; data curation, M.E.; writing—original draft preparation, M.E.; writing—review and editing, H.L. and

R.S.; visualization, M.E.; supervision, H.L.; project administration, H.L.; funding acquisition, H.L. All authors have read and agreed to the published version of the manuscript.

**Funding:** This research was funded by the University of Agder, grant number: 163831-100.

**Institutional Review Board Statement:** Not applicable.

**Informed Consent Statement:** Not applicable.

**Data Availability Statement:** Data supporting reported results are shown in the present article as tables and figures. The original data are available as Excel files, stored by the first author.

**Conflicts of Interest:** The authors declare no conflict of interest. The funders had no role in the design of the study; in the collection, analyses, or interpretation of data; in the writing of the manuscript; or in the decision to publish the results.

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
