# Peer review of "Hydroponic Lettuce Cultivation Using Organic Nutrient Solution from Aerobic Digested Aquacultural Sludge"

_agronomy, doi:10.3390/agronomy11081484_

Round 1

Reviewer 1 Report

The manuscript “Hydroponic Lettuce Cultivation Using Organic Nutrient Solution From Aerobic Digested Aquacultural Sludge” seems to be an interesting topic. The authors tried to compare the effect of processed organic nutrient solution and conventional nutrient solution on growth and quality changes in lettuce cultivation.

I appreciate the data and overall presentation of the manuscript.

However, as you analyzed the data using Tukey’s HSD test, why did not you consider using asterisks/lettering to show the mean differences (whether it is significant or not) for better understanding.

Author Response

Point 1: Why did not you consider using asterisks/lettering to show the mean differences (whether it is significant or not) for better understanding.

Response 1: As a response to the valuable comment from the reviewer, I added different letters to indicate statistically significant differences in table 4 and in figures 3, 4 and 5. Please see the attachment.

Reviewer 2 Report

This manuscript explores the possibility of using aquacultural residues as fertilizer for hydroponic cultures. For that, before use it, the sludge is processed by aerobic digestion (AD) followed by a clarification step. They characterize the product of the AD (chemical oxygen demand, total suspended solids, biological oxygen demand, total N and total P) as well as its nutrient composition (macro and micronutrients) and compare it with the sludge before the AD observing a general improvement of the residue characteristics. They use the processed aquacultural residues as growing solution for a hydroponic culture of lettuce and characterize the evolution of the solution (pH, NO3 and PO4 contain) and the final characteristics of the culture (fresh weight, macro and micronutrients contain, yield, water consumption) and the accumulation of heavy metal in the vegetable.

 The work is appropriately designed and well performed and report data and conclusions that can be interesting for the scientific community. However, there are some comments and suggestions that I would like to discuss with the authors.

The manuscript is mainly descriptive, and the discussion is a bit dull and superficial. I recommend them to discuss their results in depth. For example, lettuces grown with the organic solution consume significantly less water that those grown with the conventional nutrient solution. Do they have any hypothesis to explain this? As well, Cd level in lettuce leaves grown with their organic nutrient solution is above the maximum permissible limits and Zn and Pb are close to this limit. However, the authors conclude that this organic nutrient solution “can be used directly for lettuce production in a hydroponic system” although clarify that the concentration of heavy metals should be monitored. In my opinion, with values as close or above the maximum permissible concentration, one cannot take the risk to use this solution for fertilization or even more, I am not sure that it will be approved by the authorities. Could the authors suggest any efficient strategy or treatment to reduce the concentration of these metals in the organic solution?

Nutrient content of the organic and the conventional nutrient solution are different, which make difficult to compare their effect on the lettuce culture. Do the authors know the critical concentration of the different macro and micronutrients for lettuce culture? If so, are any of them below that limit? This could help to explain the differences between both cultures.

Some minor details:

Graphs have different format. I suggest the author to adjust all of them to the same format for the seek of clarity.

Lane 176: s-2 should be s-1

Lanes 371-372: “The ONS was higher in concentration for all these elements, except for S”. What about B?

The authors remark that the have included the outliners in the statistical analysis. Any specific reason for this?

Where the lettuce growth studies performed only once? How many plants do they use? 12? I have deduced this from the explanation of the growing units, but it may need to be explained more explicitly.

Author Response

Point 1:  Lettuces grown with the organic solution consume significantly less water that those grown with the conventional nutrient solution. Do they have any hypothesis to explain this?

Response 1: This issue has been explained on page 10 line 392.

Point 2: Cd level in lettuce leaves grown with their organic nutrient solution is above the maximum permissible limits and Zn and Pb are close to this limit. However, the authors conclude that this organic nutrient solution “can be used directly for lettuce production in a hydroponic system” although clarify that the concentration of heavy metals should be monitored.  In my opinion, with values as close or above the maximum permissible concentration, one cannot take the risk to use this solution for fertilization or even more, I am not sure that it will be approved by the authorities. Could the authors suggest any efficient strategy or treatment to reduce the concentration of these metals in the organic solution?

Response 2: We removed “can be used directly" and replaced by "can be used safely after applying proper heavy metal removal treatment". We suggested some techniques to reduce the concentration of heavy metals in page 11 line 420.

Point 3: Nutrient content of the organic and the conventional nutrient solution are different, which make difficult to compare their effect on the lettuce culture. Do the authors know the critical concentration of the different macro and micronutrients for lettuce culture? If so, are any of them below that limit? This could help to explain the differences between both cultures.

Response 3: It is normal to have different nutrient content and even conventional nutrient solutions have different nutrient concentrations depending on the producers. Lettuce can grow well in a large range of nutrient concentration and we presented the standard recommended range for each macro and micronutrient in tale 2. We mentioned in page 6 line 264 that some nutrients are below the recommended range. But this did not affect lettuce growth and yield as we mentioned. 

Point 4: Graphs have different format. I suggest the author to adjust all of them to the same format for the seek of clarity.

Response 4: Figure 2 and 3 are both consisted of two graphs (a and b). Figure 4 and 5 consisted of only one graph. If I unified the format according to figure 4, the first figures (figure 2 and 3) will be very small and unclear for the reader. And If I unified  the format according to the first figures,  figure 4 and 5 will be very stretched.

Point 5:  s-2 should be s-1 in line 176.

Response 5: We changed s-2 to s-1.

Point 6: "The ONS was higher in concentration for all these elements, except for S”. What about B?

Response 6: We added B in page 10 line 372.

Point 7: The authors remark that the have included the outliners in the statistical analysis. Any specific reason for this?

Response 7: In our case outliers are due to variability in the measurement. We included them because they have influence on  mean.

point 8: Where the lettuce growth studies performed only once? How many plants do they use? 12? I have deduced this from the explanation of the growing units, but it may need to be explained more explicitly.

Response 8: We performed the growth studies in three identical pipes with a total of 36 plants. We mentioned that in the materials and methods section "the NFT-system  consisted of three identical parallel units. Each unit consisted of a rectangular PVC-pipe (240 cm long, 10 cm width and 50 cm height) with 12 holes".  lettuce are placed in the holes: 12 * 3 = 36.